# IDA-VLM: Towards Movie Understanding via ID-Aware Large Vision-Language Model

**Yatai Ji**[1,2], **Shilong Zhang**[1], **Jie Wu**[3], **Peize Sun**[1],
**Weifeng Chen**[3], **Xuefeng Xiao**[3], **Sidi Yang**[2], **Yujiu Yang**[2]*, **Ping Luo**[1]*
[1]The University of Hong Kong,     [2]Tsinghua University,     [3]ByteDance

## Abstract

The rapid advancement of Large Vision-Language models (LVLMs) has demonstrated a spectrum of emergent capabilities. Nevertheless, current models only focus on the visual content of a single scenario, while their ability to associate instances across different scenes has not yet been explored, which is essential for understanding complex visual content, such as movies with multiple characters and intricate plots. Towards movie understanding, a critical initial step for LVLMs is to unleash the potential of character identities memory and recognition across multiple visual scenarios. To achieve the goal, we propose visual instruction tuning with ID reference and develop an **ID-A**ware Large **V**ision-**L**anguage **M**odel, IDA-VLM. Furthermore, our research introduces a novel benchmark MM-ID, to examine LVLMs on instance IDs memory and recognition across four dimensions: matching, location, question-answering, and captioning. Our findings highlight the limitations of existing LVLMs in recognizing and associating instance identities with ID reference. This paper paves the way for future artificial intelligence systems to possess multi-identity visual inputs, thereby facilitating the comprehension of complex visual narratives like movies.

## 1 Introduction

Our real world contains a wide variety of information, such as texts, images, sounds, etc. Towards multimodal comprehension (Guo et al., 2019; Lu et al., 2023), inspired by the success of Large Language Models (LLMs) (OpenAI, 2023a; vicuna, 2023; Touvron et al., 2023; Meta, 2024; Google, 2023a), there is a surging interest in Large Vision-Language Models (LVLMs), such as LLaVA (Liu et al., 2023a), Otter (Li et al., 2023a), GPT-4V (OpenAI, 2023b), Gemini (Google, 2023b), and others. In the quest for Artificial General Intelligence (AGI), LVLMs serve as pivotal milestones, enhancing machines' capabilities in multimodal perception, reasoning, and knowledge.

Typical LVLMs incorporate visual encoders with LLMs, such as LLaVA (Liu et al., 2023a), MiniGPT4 (Zhu et al., 2023). These models project an image's visual features into the embedding space of language models and employ visual instruction tuning, allowing users to complete a variety of visual tasks through language instructions, as depicted in Figure 1(a). However, pure language interaction makes VLMs hard to receive precise Region-of-Interest references from users, thus hindering their capability to focus on specific regions of an image. To address this issue, a series of works (Chen et al., 2023b; Zhang et al., 2023; Cai et al., 2023; Chen et al., 2023a; Zhao et al., 2023b; Rasheed et al., 2023) attempts to allow adding RoI references to language instructions and then use region-level data to align these RoI references to the image. As shown in Figure 1(b), these excellent works usually support using position coordinates or visual prompts as the RoI reference, which provides a more flexible interactive experience within specific areas.

Previous LVLMs have demonstrated versatile capabilities for visual understanding from the image level to Region-of-Interest. However, these models can only process the visual input of a single scenario, and their ability to associate instances across different scenes has not yet been explored, which requires the model to memorize the instance identity and recognize it in different scenes. This ability, which we call 'ID awareness', is a fundamental human visual competency essential for

---

* Corresponding author.

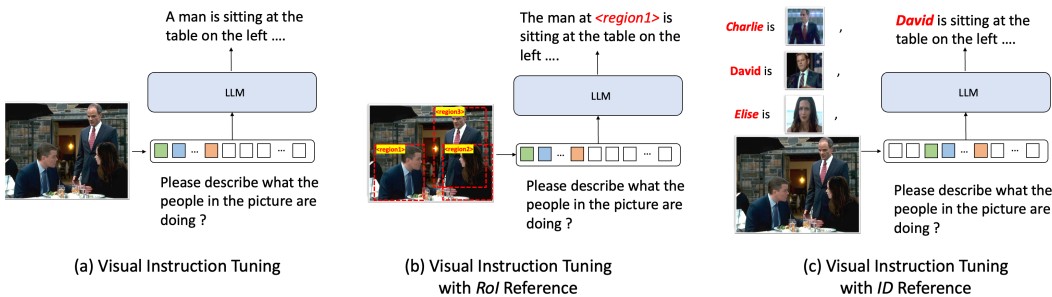

Figure 1: **Comparison of different Visual Instruction Tuning Formats.** For visual instruction tuning with ID reference, we arrange the names and images of each character as references. The model should be able to recognize the correct character identity and then answer the user's instructions.

comprehending complex multi-identity visual inputs, such as movies and animations. In the context of movie understanding, viewers should remember the name and appearance of each character, then recognize them across disparate scenes correctly, linking instances across diverse images for accurate narrative interpretation.

To achieve ID awareness in LVLMs like humans, as shown in Figure 1(c), we propose visual instruction tuning with ID reference, which allows users to utilize the name and a reference image of the character (ID image) to define the identity in the prompt, and raise questions about test images. However, there is no off-the-shelf visual tuning data with ID reference, so we meticulously craft a dual-stage instruction tuning dataset based on existing datasets. The initial phase leverages annotations in datasets such as VCR (Zellers et al., 2019), Flickr30k (Plummer et al., 2017), and RefCOCO (Kazemzadeh et al., 2014) with our data configuration strategies, to tutor the model on associating the instances of ID images with test images, where ID references are extracted from the test image. The subsequent phase utilizes MovieNet (Huang et al., 2020) to generate Q&A and caption instruction tuning data with GPT-4V (OpenAI, 2023b). The second stage data contains ID references of higher quality, which is more similar to real movie understanding. These tuning data unleash the potential of LVLMs to memorize and recognize instance identities in different scenes. Based on the visual instruction tuning with ID reference, we develop an ID-aware LVLM, called IDA-VLM, suitable for multi-identity visual comprehension. Our model learns the fine-grained identity information from ID references and generalizes to recognize characters from different scenes of test images. We also introduce a specialized component, termed ID-Former, to enhance the model's capability of recognizing character identities.

To measure ID recognition capability for movie understanding, we propose a new benchmark, namely MM-ID. Our MM-ID aims at challenging LVLMs to remember and recognize instance identities, further understanding complex visual scenes. MM-ID dissects LVLMs' competency across four progressively complex levels: matching, location, question-answering, and captioning. MM-ID comprises a collection of 585 diverse testing samples, whose instance identities source from the characters of movies, the roles of animations, and individualized animals and objects. We test both open-source models and closed-source APIs, which exposes their apparent deficiencies in identity memory and recognition, and the results are unsatisfactory even for GPT-4V. The evaluation results not only reveal the current limitations of LVLMs in instance recognition with ID reference but also highlight our benchmark's significance. In contrast, IDA-VLM achieves the best performance on MM-ID, which demonstrates the effectiveness of our instruction tuning strategies and model design.

In a nutshell, our contributions are summarized as follows:

- This paper is the first attempt to investigate ID awareness of LVLMs for complex multi-identity visual recognition and comprehension, which is a critical challenge towards movie understanding. We propose visual instruction tuning with ID reference and construct corresponding tuning datasets.

- We develop an ID-aware LVLM, IDA-VLM to recognize character identities and understand visual content in an end-to-end manner. We adopt a dual-stage fine-tuning method to unleash the potential of LVLMs in identity memory and recognition across diverse scenes.

- We propose MM-ID, a novel benchmark to examine LVLMs on identity memory and recognition. We evaluate representative LVLMs on our benchmark, uncovering the limitations of existing LVLMs in contextual identity recognition. Furthermore, our IDA-VLM can serve as a reference and inspirational baseline model on MM-ID.

## 2 RELATED WORK

### 2.1 LARGE VISION-LANGUAGE MODELS

Conventional multimodal models consist of uni-modal encoders and cross-modal fusion encoders. Relying on vision-language pre-training (Tan & Bansal, 2019; Lu et al., 2019; Dou et al., 2022; Wang et al., 2021; Ji et al., 2023b), they have shown an impressive cross-modal semantic alignment ability (Ji et al., 2023a; Tu et al., 2023), which brings substantial advances on various downstream tasks (Goyal et al., 2019; Plummer et al., 2017; Lin et al., 2014). However, due to the limitations of model size and training data scale, the performance of vision-language pre-training models is unsatisfied in open-ended scenarios.

Nowadays, Large language model (LLM) has exhibited remarkable abilities to understand, reason, and generate texts. Large Vision-Language Model (LVLM) (Chen et al., 2023c; Dai et al., 2023; Dong et al., 2024) incorporates visual encoder and LLM, aligning visual features to the embedding space of LLM. Leveraging strong generalization and emergent capability of LLM, LVLM realizes free multimodal interactions with human. Flamingo (Alayrac et al., 2022) is a pioneering work on extending LLMs to vision-language pretraining by inserting additional cross-attention layers for visual input. BLIP-2 (Li et al., 2023d) proposes Q-former to map the visual features to the hidden space of language models. To date, various works have shown encouraging progress with instruction tuning, including MiniGPT-4 (Zhu et al., 2023), LLaVA (Liu et al., 2023a), Otter (Li et al., 2023a), which demonstrate impressive results on natural instruction-following and visual reasoning capabilities. The perceptual capabilities of LVLMs are evolving towards fine-grained understanding. Numerous models (Zhang et al., 2023; Liu et al., 2023a; Chen et al., 2023b; Wang et al., 2023) focus on region-level understanding, using visual instruction tuning with RoI reference, so that users could ask questions about specific instances within the content. Moreover, there are some LVLMs equiped with Stable Diffusion, which can produce multimodal outputs (Koh et al., 2023; Ge et al., 2023; Sun et al., 2023; Li et al., 2023c). LVLMs hold the potential to perform a wider array of functions and to understand more intricate visual information. For instance, movie is considered as one of the most intricate mediums for conveying visual information. This paper researches on recognizing characters across disparate scenes with ID reference, towards movie content understanding.

### 2.2 MULTIMODAL BENCHMARK

Previous evaluation for multimodal models measures some specific abilities, such as VQAv2 (Goyal et al., 2019), GQA (Hudson & Manning, 2019) for visual question answering, Ref-COCO (Kazemzadeh et al., 2014) for visual grounding, Visual7W (Zhu et al., 2016) for PointQA, VCR (Zellers et al., 2019) for commonsense reasoning in an image, ChartMimic (Shi et al., 2024) for chart understanding. Recently, there is a surging interest in developing comprehensive benchmarks for evaluating LVLMs. MMBench (Liu et al., 2023b) consists of multiple-choice questions and introduces a CircularEval strategy for evaluation framework. LLaVA-Bench (Liu et al., 2023a) employs GPT-4 to assess responses from both GPT-4 and the model under evaluation, then provides scores and explanations for the answers. MM-Vet (Yu et al., 2023) assesses LVLMs across six fundamental visual-linguistic capabilities, utilizing a GPT-4-based evaluator for open-ended responses. MME (Fu et al., 2023) tests for perception and cognition competencies across 14 distinct subtasks, which requires models to provide simple 'yes' or 'no' answers. SEED-Bench (Li et al., 2023b) covers 12 evaluation dimensions and adopts an answer ranking strategy to evaluate LVLMs via multiple-choice questions. POPE (Li et al., 2023e) is a dedicated benchmark for assessing object hallucination. Existing benchmarks evaluate multifaceted performance of LVLMs, yet they predominantly focus on visual scenarios that lack character identities and intricate plots.

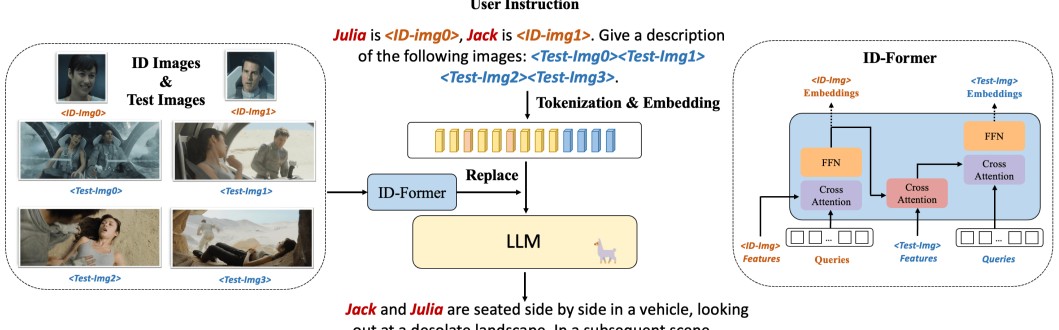

Figure 2: IDA-VLM is an end-to-end vision-language model for processing instructions that contain references to specific instances. We introduce ID reference as using a correlating character reference image and corresponding name to characterize an identity, exemplified as *Julia is <ID-Img{i}>*. During tokenization and conversion to embeddings, the embedding of *<ID-Img{i}>* and *<Test-Img{i}>* in the instruction are replaced with the ID and Test image embeddings respectively. A simple yet effective image feature projector termed ID-Former is proposed to enhance the ID identification ability. As the output in the figure, IDA-VLM can memorize these character IDs, recognize them in test images, and respond to user instructions with the correct ID references.

## 3 METHOD

Movie understanding represents a considerable departure from conventional video understanding, which typically limits to a single scenario or activity. To understand complex visual input, for example, a movie or animation, the model need memorize and recognize character identities, linking characters in distinct scenes, with each scene represented by an image in our setting. As shown in Figure 2, we simplify movie understanding as given ID images of certain characters and keyframes from the movie or animation served as test images, the model completes instruction tasks. The ID images, test images, and instruction text are sent to the model together for inference in an interleaved format of images and text. The model need to memorize instance identities in ID references and recognize them in test images for providing responses.

### 3.1 MODEL ARCHITECTURE

In this paper, we adopt Qwen-VL-Chat (Bai et al., 2023) as our baseline model. In order to adapt the model to the task of instance ID recognition, we employ a dual-stage visual instruction tuning with ID reference. The architecture of IDA-VLM comprises three components: a visual encoder, ID-Former consisting of cross-attention mechanisms, and a subsequent large language model.

As shown in Figure 2, the ID-Former is designed to project visual features into the input semantic space of LLM and contribute to recognizing instance identities. This is achieved by two cross-attention modules. The first one interacts learnable queries with the visual features through cross-attention, effectively compressing the visual semantics into a shorter, fixed-length feature encoding. The second cross-attention utilizes queries of ID images to modulate test image embeddings, activating identity information of test images.

### 3.2 FIRST-STAGE TUNING

In the first phase, we harness the off-the-shelf annotations available in existing datasets along with our proposed data configuration strategies, reducing the cost for extra annotations about ID recognition. Specifically, we utilize the public datasets containing instance spatial information, including Visual Commonsense Reasoning (VCR) (Zellers et al., 2019), RefCOCO (Kazemzadeh et al., 2014), and Flickr30k (Plummer et al., 2017) datasets, to construct visual instruction tuning data with ID reference, which contains three instruction tasks: question-answering (Q&A), location, captioning.

VCR dataset consists of questions that require commonsense reasoning to answer, as well as choosing the reasons. These questions ask about particular people or objects, which have location anno-

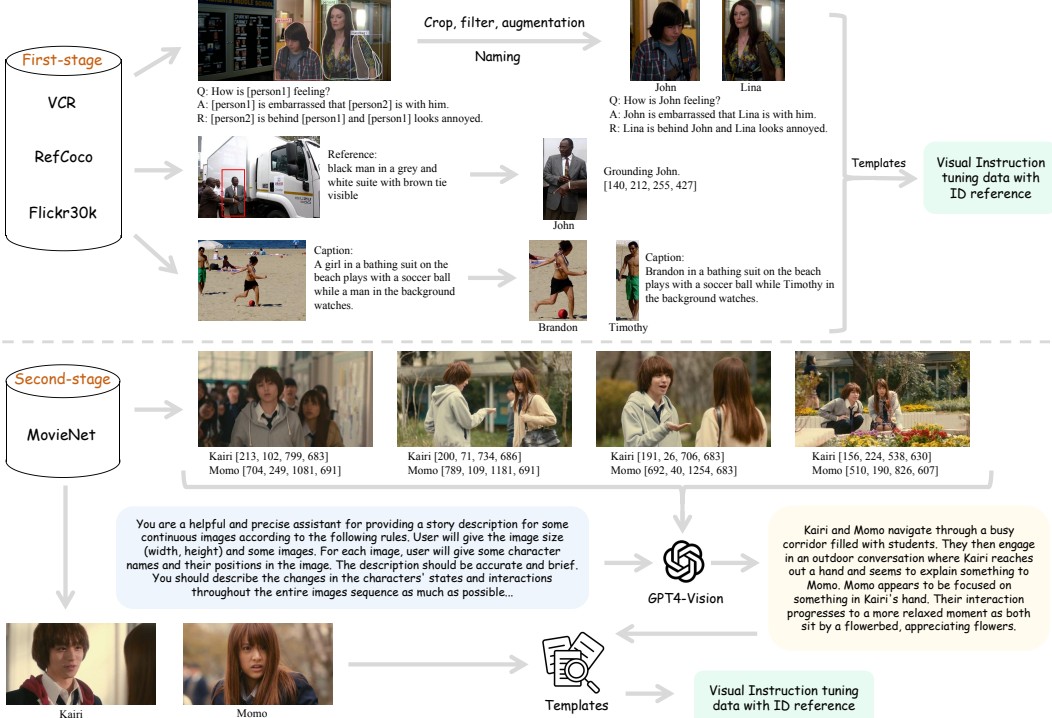

Figure 3: First-stage and second-stage instruction tuning data construction pipelines.

tations. We crop sub-images of the individuals and assign them identifiable labels, such as person names. By replacing the people in the questions and answers with these labels, we gain a new set of QA data that necessitates ID recognition. RefCOCO (and variants RefCOCO+ and RefCOCOg) is a dataset targeted on visual localization based on descriptions. For the items to be localized, we create ID images after cropping them from the original images to perform localization tasks. Flickr30k, a dataset comprising image descriptions, includes coordinates for the people or objects mentioned in the descriptions. We extract the areas associated with individuals to produce ID images and provide names to generate corresponding descriptions. Additionally, we conduct some data augmentation and cleaning operations, such as filtering out samples with only one person or those with sub-images of inappropriate sizes. The annotation pipeline is shown in the upper part of Figure 3.

The essential purpose of this stage is to train the model on its ability to build relationships between multiple images, using the ID images to identify and accurately locate or describe objects within test images. However, the instance ID images are cropped from the original images, which reduces the difficulty of recognition.

## 3.3 SECOND-STAGE TUNING

The second-stage fine-tuning data is based on the MovieNet (Huang et al., 2020) dataset. The original annotations in the MovieNet dataset encompass the names of the characters present in each movie shot and their coordinate location information. In the MovieNet, we select pictures that contain only a single character to serve as ID images, while those containing multiple characters are used as test images. This approach allows us to naturally develop a dataset for location task. Moreover, we compile ID images featuring the same character to construct data for matching task.

For Q&A and caption tasks, we adopt GPT-4V to convert annotations from the MovieNet dataset into the question format that refers to specific roles. Specifically, as illustrated in the lower part of Figure 3, we feed test images along with their character location information into GPT-4V, and encourage the model to generate captions or question-answer pairs via prompt engineering. The detailed prompts used for generating descriptions or question-answer pairs are shown in Appendix

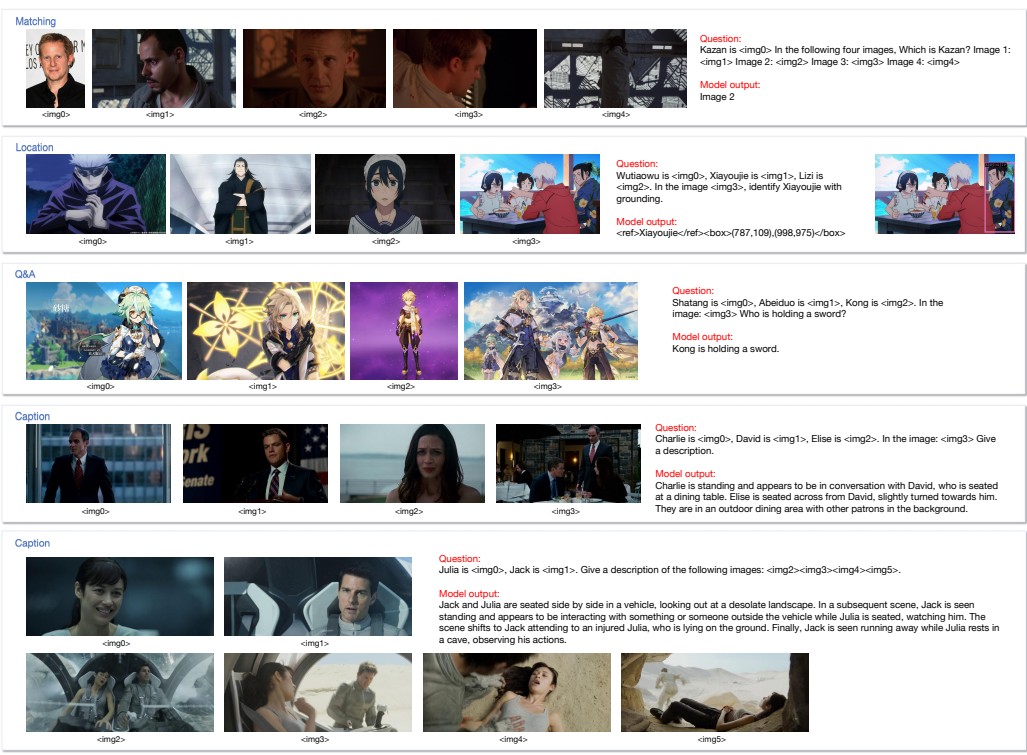

Figure 4: Data samples of MM-ID. Each sample consists of ID images of each character, and test images containing multiple characters. Our model can memorize the identity information in ID reference and generalizes to recognize characters from different scenes.

B. Finally, we integrate ID images, test images and results of GPT-4V into conversation templates, producing second-stage instruction tuning data.

After completing data construction process, we train our model with next-token prediction loss in two stages. The first stage tuning data contains approximately 80,000 samples, while the second one comprises around 60,000 samples. Moreover, to ensure that the model would not forget its previously learned knowledge during ID recognition training, we integrate instruction tuning data from LLaVA and ShareGPT4V with the aforementioned data, training them simultaneously.

# 4 MM-ID

## 4.1 Problem Definition

To measure ID recognition capability of models, we propose a new benchmark, MM-ID. We appraise the capability of models to recognize IDs across four incrementally complex levels. The first sub-task we investigate is matching: given an instance image, which could feature a person, an animal, or a building, the model must choose an image from four options that contains the same instance. The second sub-task involves localizing the instance within a test image based on the ID image, and providing coordinates of the bounding box, with the challenge coming from numerous similar distractor objects or individuals present in the test image. The third and fourth sub-tasks—question-answering and caption generation—require the model to recognize the identities in the test images, then either respond to related questions or produce a comprehensive description. These represents the most difficult level of tasks, as the model should not only recognize each instance's identity in the image but also provide appropriate answers based on information related to their names, including states, actions, locations, and more. The samples of sub-tasks are shown in Figure 4.

## 4.2 Data Annotation and Statistics

In the construction of MM-ID benchmark, we initially collect images that meet the specified conditions. To evaluate the model's capability of recognizing instance-level information, the images used for testing need to contain multiple characters. Furthermore, each character requires distinctiveness, such as different actions, clothing attributes, etc., to facilitate the design of questions. Our data sources primarily encompass three types: shot images of movie scenes from the MovieNet dataset, animated images with download links collected from the internet, and pictures of buildings, vehicles, and animals gathered from object re-identification (Re-ID) datasets (Ye et al., 2024; Jiao et al., 2023). In each example, there exists ID images corresponding to each character and one or more test images for inquiry.

Following preparing images of MM-ID, we annotate questions and standard answers manually. For description and localization sub-tasks, we use GPT-4 to generate question templates. In question-answering task, annotators pose questions regarding specific characters' actions, attributes, clothing colors, positions, relationships, emotions, etc., following detailed annotation documents and guidance examples. The question design ensures that the model can only provide a correct answer when it accurately recognizes the characters. After question annotation, we request other individuals' assistance in refinement and balancing, increasing the diversity of expression and grammatical structure. Finally, the correct answers are labeled manually by other four annotators, providing accurate and detailed descriptions for caption task and correct answers for Q&A task. Everyone participates in discussions and makes peer reviews for each other. The instructions for question annotations and quantitative evaluation of inter-annotator agreement are shown in Appendix C.

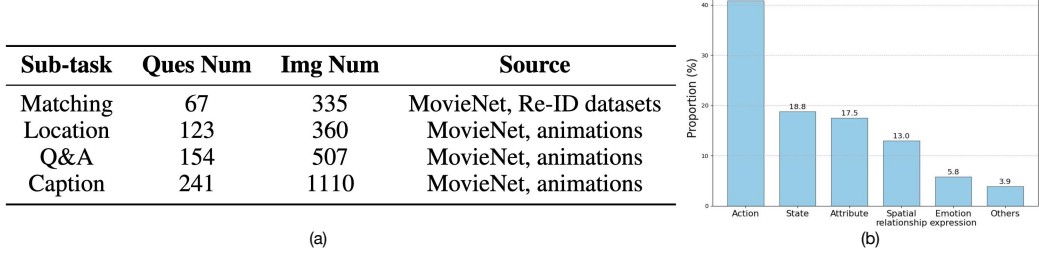

| Sub-task | Ques Num | Img Num | Source |
|----------|----------|---------|--------|
| Matching | 67 | 335 | MovieNet, Re-ID datasets |
| Location | 123 | 360 | MovieNet, animations |
| Q&A | 154 | 507 | MovieNet, animations |
| Caption | 241 | 1110 | MovieNet, animations |

(a)  (b)

Figure 5: MM-ID data statistics. (a) Sample number and data sources of four sub-tasks. (b) The proportion of each perspective of Q&A.

Totally, we collect and annotate 585 samples. The data composition is shown in Figure 5 (a). The "Q&A" and "Caption" sub-tasks are main parts of MM-ID benchmark. Because we use GPT-4[1] for scoring, it is costly to have too many evaluation samples and we select the most representative samples for MM-ID. As shown in Figure 5 (b), Questions of "Q&A" task comprehensively covers different perspectives of characters (actions, states, attributes, positions, emotions), which can sufficiently evaluate the ID recognition capability of LVLMs.

## 4.3 Evaluation Strategy

Quantitative evaluation for open-domain LVLMs has always been challenging. Our MM-ID incorporates four sub-tasks, each presenting its unique answer format. For matching task, which resembles multiple-choice questions, accuracy is used as a metric. In localization task, we compare the predicted bounding box to the actual bounding box, computing the Intersection over Union (IoU) metric to measure accuracy. When IoU exceeds a threshold of 0.5, we consider the model recognizes instance identity accurately.

For question-answering and caption generation tasks, due to the open-ended nature of the generated responses, it is challenging to employ rule-based evaluations. Hence, inspired by MM-Vet and LLaVA-Bench, GPT-4 is used to score the results. Under the condition of provided questions and correct answers, we design prompts to guide GPT-4 to focus on correspondence of character names and their states or actions, scoring the accuracy of characters' descriptions within model predictions.

---

[1] version: gpt-4-1106-preview

Table 1: The comparison results on MM-ID. Notably, The scores of 'Q&A' and 'Caption' are evaluated with GPT-4 absolute scoring strategy. '-' indicates the model can't complete corresponding instruction.

| Model | Matching | Location | Q&A | Caption |
|---|---|---|---|---|
| **Open-source Models** | | | | |
| MMICL (Zhao et al., 2023a) | – | – | 3.53 | 3.18 |
| SEED (Ge et al., 2023) | – | – | 3.19 | 3.58 |
| LLaVA-OneVision (Li et al., 2024) | – | – | 3.91 | 2.86 |
| Qwen-VL-Chat (Bai et al., 2023) | – | 0.504 | 3.63 | 2.65 |
| InternLM-XComposer2 (Dong et al., 2024) | – | 0.106 | 3.44 | 2.93 |
| **Closed-source APIs** | | | | |
| Qwen-VL-Plus (Bai et al., 2023) | 0.313 | 0.187 | 3.87 | 3.79 |
| Qwen-VL-Max (Bai et al., 2023) | 0.224 | 0.301 | 4.64 | 4.23 |
| Gemini-pro (Google, 2023b) | 0.687 | 0.081 | 4.97 | 4.04 |
| GPT-4V (OpenAI, 2023b) | 0.627 | 0.244 | 4.77 | 4.67 |
| IDA-VLM | **0.716** | **0.821** | **5.71** | **5.15** |

It is found that outputs of GPT-4 still exist variance, so we utilize GPT-4 to score the predictions by 5 times and report average scores.

We use both absolute and relative scoring strategies. Absolute scoring rates a model's prediction directly against the correct answer on a ten-point scale, while relative scoring pits two models' results against each other, providing more immediate comparative insights into the models' performance with a fraction. The prompts used for GPT-4 scoring are shown in Appendix C. Besides using GPT-4 scoring, we also employ conventional caption metrics like METEOR and CIDEr.

## 5 EXPERIMENTS

### 5.1 QUANTITATIVE RESULTS

We use Qwen-VL-Chat for model initialization, which has 9.6B parameters. During IDA-VLM training, we set learning rate as 1e-5 for the first stage and 5e-6 for the second stage. The model is trained for 5 epochs in both the first and second stages. The batch size with gradient accumulation is set to 128. The visual encoder is fixed, while ID-Former and LLM are fine-tuned.

We compare IDA-VLM with other open-source LVLMs (Zhao et al., 2023a; Ge et al., 2023; Bai et al., 2023; Dong et al., 2024) and closed-source APIs (Bai et al., 2023; Google, 2023b; OpenAI, 2023b) on MM-ID. To meet the requirement of ID recognition, the models selected for testing should support multiple images input. Because they are not fine-tuned for this task, the prompt is crucial for guiding models on ID awareness. We design appropriate prompts to guide them to respond according to character names. The details can be found in Appendix D.

As illustrated in Table 1, our model achieves the best performance on MM-ID. According to the quantitative results, Gemini-pro exhibits best performance on matching and Q&A sub-tasks among previous models. In the location sub-task, Qwen-VL-Chat gains the highest score, even higher than other closed-source APIs. As for the caption sub-task, GPT-4V surpasses other models. Overall, our model outperforms previous LVLMs by a significant margin on four sub-tasks. The results demonstrate visual instruction tuning with ID reference unleashes the potential of LVLM on ID awareness, which can not be realized only by prompt engineering.

As shown in Table 2, we calculate relative scores using GPT-4 to provide a further demonstration on the 'Q&A' and 'Caption' sub-tasks. We compare the predictions from various closed-source APIs against those from our model respectively. All relative scores are below 1, signifying that our model outperforms its counterparts. Among the closed-source APIs evaluated, GPT-4V exhibits comparatively superior results. In Table 3, our model also gains best performance on traditional metrics of caption task.

Table 2: Using relative scoring strategy to compare previous LVLMs with our model. Each result is the ratio of the first model's score to that of the second model's, after feeding the predictions of two models into GPT-4 for comparison.

| Model | Q&A | Caption |
|---|---|---|
| GPT-4V / IDA-VLM | 0.925 | 0.793 |
| Gemini-pro / IDA-VLM | 0.892 | 0.704 |
| Qwen-VL-Chat / IDA-VLM | 0.765 | 0.396 |
| Qwen-VL-Plus / IDA-VLM | 0.796 | 0.544 |
| Qwen-VL-Max / IDA-VLM | 0.918 | 0.636 |

Table 3: METEOR and CIDEr metrics evaluation on 'Caption' sub-task. Both metrics take into account the semantic similarity of the captions.

| Model | METEOR | CIDEr |
|---|---|---|
| Qwen-VL-Chat | 0.052 | 0.031 |
| Qwen-VL-Plus | 0.119 | 0.140 |
| Qwen-VL-Max | 0.134 | 0.083 |
| Gemini-pro | 0.121 | 0.185 |
| GPT-4V | 0.144 | 0.078 |
| IDA-VLM | **0.191** | **0.515** |

Table 4: Performance of IDA-VLM on standard benchmarks. "w/o LLaVA data" means finetuning with only constructed ID reference datasets.

| Model | MME | | SEED-Bench | |
|---|---|---|---|---|
| | Perception | Cognition | Image | Video |
| Qwen-VL-Chat | 1441.4 | **387.5** | **0.650** | 0.397 |
| IDA-VLM (w/o LLaVA data) | 1167.5 | 349.6 | 0.441 | 0.311 |
| IDA-VLM | **1484.6** | 383.6 | 0.647 | **0.412** |

To assess the basic multimodal capabilities, we also evaluate IDA-VLM with the baseline model Qwen-VL-Chat on standard benchmarks. As shown in Table 4, the perception metric on MME gets improved, which is due to the ID-level recognition learning boosts fine-grained perception ability. On SEED-Bench, our training samples containing multiple test images enhance video understanding ability. Comparing the last two rows, the inclusion of LLaVA and ShareGPT4V data greatly contributes to preserving the original capabilities.

## 5.2 QUALITATIVE COMPARISON

As shown in Figure 6, we qualitatively compare the results of IDA-VLM with other models. In the first case, GPT-4V does not exhibit accurate localization ability, and Qwen-VL-Chat mistakenly recognizes Leizi as another similar person. In the second case, the model should determine which person walks in front and match the person with the given characters. Our model gives the right answer 'Ariadne' while GPT-4V and Gemini-pro recognizes inaccurately. In the last case, models are asked to provide a description for a movie scene with three characters. GPT-4V describes that David and Elise sit at a table, ignoring Charlie standing. Gemini-pro only gives a generic caption for the test image without recognizing characters' identities. In contrast, our model gives the most reasonable description of the scene. We present more qualitative examples in Appendix E.

## 5.3 ABLATION STUDIES

**Effect of Dual-stage instruction tuning.** We report the separate effect of each instruction tuning stage in Table 5. When the second stage of tuning is removed, there is a significant drop in performance, suggesting that the second stage has a more substantial impact than the first. During the second stage, the ID images are independent of the test images, which enhances the quality of data for model learning. When the first and second stages of tuning are combined, our model achieves its optimal performance. It is worth noting that the accuracy of 'Matching' gets lower when adding the first stage tuning, because the first stage tuning data can't improve matching ability.

Table 5: The ablation study about separate stage instruction tuning.[1]

| Model | Matching | Location | Q&A | Caption |
|---|---|---|---|---|
| w/o first stage tuning | **0.746** | 0.813 | 5.43 | 5.08 |
| w/o second stage tuning | – | 0.715 | 4.08 | 3.86 |
| Ours | 0.716 | **0.821** | **5.80** | **5.19** |

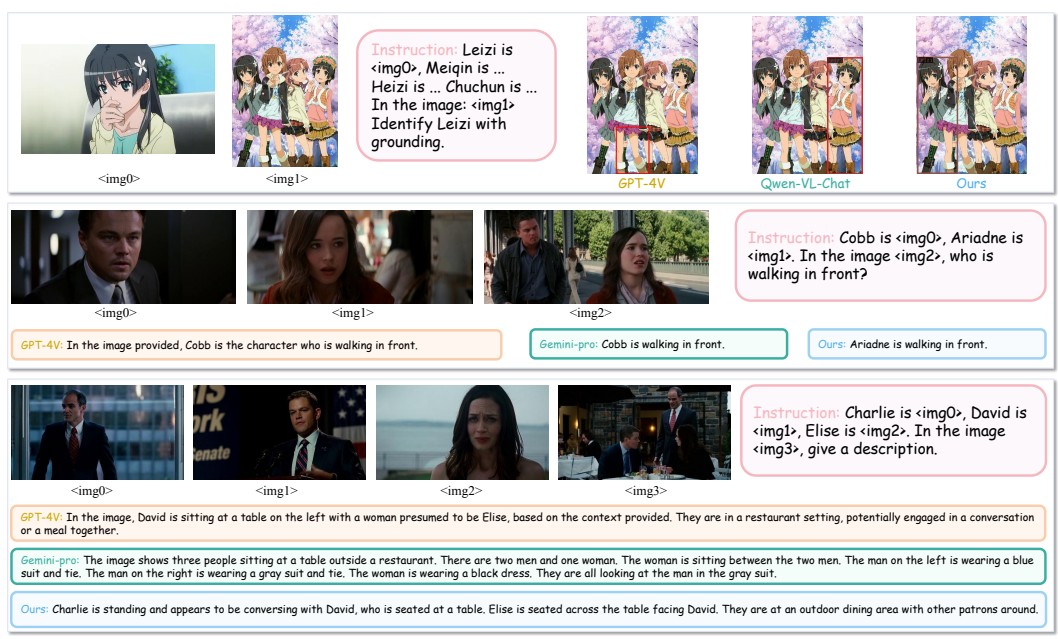

Figure 6: We present visualizations of selected samples from MM-ID, corresponding to location, Q&A and caption sub-tasks. We showcase results of some representative models and ours (IDA-VLM). The detailed prompts for other models to complete instructions are shown in Appendix D.

Table 6: The ablation study about ID-Former.

| Model | Matching | Location | Q&A | Caption |
|---|---|---|---|---|
| w/o ID-Former | 0.701 | 0.803 | 5.44 | 5.13 |
| Ours | **0.716** | **0.821** | **5.71** | **5.15** |

**Effect of ID-Former.** We use ID-Former to project visual features of ID images and test images to the semantic space of LLM. As depicted in Table 6, substituting the ID-Former with a standard query former leads to a reduction in all sub-task metrics. The ID-Former enhances the interaction between features of ID images and test images, thereby activating ID information within the test images.

The ablation study for the mixing rate of LLaVA and ShareGPT4V instruction tuning data are depicted in Appendix B.

## 6  CONCLUSION

In this paper, we focus on the capability to recognize and link instances across various scenes, which is significant for understanding complex visual narratives, such as movies. We propose visual instruction tuning with ID reference, which unleashes the potential of LVLM in ID recognition and develop an ID-aware LVLM, IDA-VLM. The model memorizes the name and appearance of each character in ID reference, then recognizes them across disparate scenes correctly, and understands visual narrative in an end-to-end manner. To thoroughly assess ID awareness of LVLMs, a novel benchmark named MM-ID has been introduced, which consists of four sub-tasks. Our model can serve as a reference and inspirational baseline, which achieves best performance among previous models and closed-source APIs. Conclusively, this research contributes to broadening the horizons for future AI systems to efficiently understand multi-identity visual content.

---

[1]Because the sample of the first stage tuning has only one text image, we present the 'QA' and 'caption' results on a sub-set of MM-ID, which has one test image in each sample.

## 7 ACKNOWLEDGMENTS

This work was partly supported by the National Key Research and Development Program of China (No. 2024YFB2808903) , the research fund of Tsinghua University - Tencent Joint Laboratory for Internet Innovation Technology. This paper is partially supported by the National Key R&D Program of China No.2022ZD0161000 and the General Research Fund of Hong Kong No.17200622 and 17209324.

## 8 LEGAL STATEMENT

This work utilized the MovieNet dataset, which is publicly available under open-source license for academic research. For animation images from the internet, we provided web urls of the original content rather than redistributing the images directly, to respect potential copyright restrictions. The authors acknowledge full responsibility for the usage of all datasets in this research and have made best efforts to comply with relevant terms of use and copyright laws. The authors are responsible for any legal violations of data rights.

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
