## A    LIMITATIONS AND ETHICAL CONSIDERATIONS

As we employ Qwen-VL-chat as the baseline model for our research, our proposed model inherits certain limitations thereof. One such constraint is the upper limit on the number of input images. Within a sample, the sum count of ID images and test images can not surpass eight. Consequently, we opt to use keyframes as test images to effectively perform comprehension of movie segments. By integrating our visual instruction tuning with ID reference, if we fortify the baseline, which is anticipated to enhance overall performance.

Our method is related to person and instance ID, so we pay much attention when collecting data. Our tuning data and benchmark data mainly come from MovieNet, which is a open-source movie dataset, containing numerous shot images of actors, necessitating careful consideration about the copyright. Moreover, there exists the potential risk that our model or dataset may be utilized by others to engage in activities concerning a particular individual.

## B    VISUAL INSTRUCTION TUNING DATA CONSTRUCTION

We construct dual-stage instruction tuning data with ID reference. The first stage data is constructed with predefined rules, cropping specific instances from test images, while the second stage data is generated by GPT-4V. Specifically, GPT-4V is utilized to produce four kinds of tuning datasets: single-image caption, multi-image caption, question-answer pair for a single image, and question-answer pair for multiple images. We present prompts for producing data in Table 7 and Table B. The constructed tuning data scale of two stages are shown in Table 9.

To preserve the innate proficiency of the baseline model, we integrate the instruction tuning dataset from LLaVA and ShareGPT4V with our compiled dataset. To simplify, we refer to these tuning data as the LLaVA dataset. As depicted in Table 10, we conduct an ablation study on the proportion of LLaVA dataset. The capability for 'Matching' reaches its best in the absence of LLaVA dataset, suggesting LLaVA data has no benefit for 'Matching'. The location score is highest at a 20% inclusion rate of LLaVA data. In contrast, 'Q&A' and 'Caption' scores reach their optimal levels when the inclusion rate is set at 10%. This could imply that a judicious amount of LLaVA data serves to fortify foundational multimodal and text generation competencies. However, an excessive infusion of LLaVA data seems to hinder the learning of ID recognition. Therefore, we select 10% as the mixing rate of LLaVA data.

## C    MM-ID

During MM-ID construction, we annotate questions and standard answers manually. The instructions for annotators are shown in Table C. We have four annotators for answer annotations. We conduct a quantitative evaluation of inter-annotator agreement on 'Q&A' and 'Caption' sub-tasks, which need annotated answers. CLIP text embeddings are used to measure the similarities of answers from four annotators. We calculate mean value of each pair's similarity in four annotated answers. The agreement value distribution is shown in Table 12 and 13.

It is a challenge to evaluate accuracy of the model on 'Q&A' and 'Caption' sub-tasks. We utilize GPT-4 to score the predictions under the condition of provided questions and correct answers. We propose GPT-4 absolute and relative scoring strategies with designed prompts, as depicted in Table C. We sample our model's answers on 100 MM-ID samples and score them both manually and using GPT-4. The correlation between the two sets of scores was very high, reaching 0.83 on Spearman coefficient.

The calculate process of relative score is as follows: GPT-4 gives two scores (10 points) for predictions from different models respectively. We compute mean value of scores for each model, then calculate the ratio of the average scores as the final relative score.

## D    EXPERIMENT SETTINGS

To promote the evaluation of models on MM-ID tasks, we craft specific prompts to instruct models in accurately identifying character IDs. The prompts tailored for closed-source APIs are detailed

Table 7: Prompts for GPT-4v to annotate 'Caption' instruction tuning data based on MovieNet.

**Description generation for a single test image**
You are a helpful and precise assistant for providing a description for an appropriate image. User will give an image and the image size (width, height). Then user will give some character names and their position in the image. The position is expressed with bounding box, which is the person's left-top corner coordinates and right-bottom corner coordinates (left, top, right, bottom).
Firstly, you need to judge if it is appropriate. An appropriate image should be clear, should be easy for you to give a caption, the people in it should be easy to recognize.
If the image is not appropriate, you should answer 'no', if it is appropriate, you should give an accurate description of the image with given character names according to the following rules.
Different characters will be split by '\n', you must remember the right people in the right position. Maybe there are some other people without name in the image, your caption need to contain them if necessary, but the main subject should be about characters with names.
The description should be accurate and brief. The answer should be less than 60 words. Please pay more attention to people's action. Your answer should be only about the visual content, don't include your own speculation.
Then you should give an accurate description of the image with given character names.

**Description generation for multiple test images**
You are a helpful and precise assistant for providing a description for some continuous images. You can treat it as video caption generation. You need give an overall story description for these images according to the following rules.
User will give the image size (width, height) and some images. For each image, user will give some character names and their positions in the image. The position is expressed with bounding box, which is the person left-top corner coordinates and right-bottom corner coordinates (left, top, right, bottom).
Different characters will be split by '\n', you must remember the right people in the right position. Maybe there are some other people without name in the image, your caption need contain them if necessary, but the main subject should be about characters with names.
The description should be accurate and brief. The answer should be less than 60 words. Please pay more attention to people's action. Your answer should be in accordance with temporal order of the images. Your description should be coherent and have some logical connections. Your answer should be only about the visual content, don't include your own speculation.
You should describe the changes in the characters' states and interactions throughout the entire images sequence as much as possible, avoiding fragmented descriptions for each individual image. Especially refrain from using phrases like 'in the image 1' and so on.
Then you should give an accurate and brief description of the images with given character names.
A good example: 'Carol watches as Hal, in a white tank top, looks at his shirt. As she approaches, they engage in a close conversation, and eventually, Hal looks at Carol while gesturing, continuing their discussion.'

Table 8: Prompts for GPT-4v to annotate 'Q&A' instruction tuning data based on MovieNet.

**Q&A generation for a single test image**

You are a helpful and precise assistant for providing a question-answer pair of an image with given character names.

User will give an image and the image size (width, height). Then user will give some character names and their position in the image. The position is expressed with bounding box, which is the person left-top corner coordinates and right-bottom corner coordinates (left, top, right, bottom).

Firstly, you need to judge if it is appropriate. An appropriate image should be clear, should be easy for you to give a caption, the people in it should be easy to recognize.

If the image is not appropriate, you should answer 'no', if it is appropriate, you should give a question-answer pair of the image with given character names according to following rules.

Different characters will be split by '\n', you must remember the right people in the right position.

Then you should give a pair of question and corresponding answer about the image with given character names. The question and answer should be split by '\n'.

The question asks about the given character, including character actions, character attributes (clothes, expression, etc), character locations, relative relationship between characters, etc. Only include questions that have definite answers. Some examples of question templates: What is xxx doing? What color is xxx's clothes?

The question and answer should be accurate and brief. The answer should be strictly correspond to the question and be less than 30 words.

**Q&A generation for multiple test images**

You are a helpful and precise assistant for providing a question-answer pair for some continuous images with given character names. You can treat it as video queation answering generation. You need give an overall question and answer for these images according to the following rules.

User will give the image size (width, height) and some images. For each image, user will give some character names and their positions in the image. The position is expressed with bounding box, which is the person left-top corner coordinates and right-bottom corner coordinates (left, top, right, bottom).

Different characters will be split by '\n', you must remember the right people in the right position.

Then you should give a pair of question and corresponding answer about the images with given character names. The question and answer should be split by '\n'.

The question asks about one of or some given characters, including character actions, character attributes (clothes, expression, etc), relative relationship between characters, etc. Only include questions that have definite answers.

The question and answer should be accurate and brief. The answer should be strictly correspond to the question and be less than 30 words.

You should focus on the changes in the characters' states, actions or interactions throughout the entire images sequence as much as possible, avoiding fragmented question for each individual image. Especially refrain from using phrases like 'in the image 1' and so on.

A good example: 'What is Timmy doing?\nTimmy walks into the room, then has a conversation with another man, finally they hug each other excitedly.'

Table 9: ID recognition instruct-tuning dataset.

| Stage | tuning data | data scale |
|---|---|---|
| The first stage | VCR | 30k |
| | RefCoco | 20k |
| | Flickr30k | 30k |
| The second stage | Matching | 4k |
| | Location | 12k |
| | Single-image Q&A | 15k |
| | Single-image Caption | 15k |
| | Multi-image Q&A | 4k |
| | Multi-image Caption | 10k |

Table 10: The ablation study about instruction tuning data of LLaVA.

| Model | Matching | Location | Q&A | Caption |
|---|---|---|---|---|
| Ours (w/o LLaVA data) | **0.746** | 0.797 | 5.27 | 4.98 |
| Ours (10% LLaVA data) | 0.716 | 0.821 | **5.71** | **5.15** |
| Ours (20% LLaVA data) | 0.716 | **0.829** | 5.67 | 5.11 |

in Table D. and prompts for open-source models are designed similarly. We try multiple prompts for models to complete instructions, but none achieves satisfactory results, which demonstrates ID awareness can not be unleashed only by prompt engineering.

## E    MORE VISUALIZATION

In this section, we present more qualitative results of IDA-VLM, as shown in Figure 7, Figure 8, Figure 9, Figure 10.

## F    INFLUENCE OF ID IMAGE NUMBER

In actual movie understanding, characters of interest may not be in the test images, so the model's performance in scenarios with additional reference IDs is worth an investigation. In some samples of MM-ID, we give more ID images than characters appearing in the test image. As shown in Figure 11, in some easy Q&A samples, IDA-VLM can give right answers, but in caption sub-task, IDA-VLM may generate content of non-existing characters. This is bacause in our training data, the number of ID images generally equals to the character count in test images. If we add additional ID images to the training samples, this issue can be resolved.

## G    ROBUSTNESS TO ID VARIATIONS

In our test samples, the ID images are clear, enabling the model to capture complete identity characteristics. The test images exhibit various ID variations, including pose and clothing changes, demonstrating our model's robustness to identity variations. As shown in Figure 12, we showcase model performance under different ID variations. We investigate ID variations across three aspects: clothing, viewing angle, and lighting. The model demonstrates the highest robustness to clothing variations. For the other two aspects, we present one successful case and one failure case each. The model's performance degrades when encountering substantial changes in viewing angles, presence of occlusions, or poor lighting conditions. Moreover, the primary challenge arises from interference by IDs with similar features in the test images.

Table 11: The instructions for Annotators.

> Design Questions: The questions are about specific characters in one or more images. These questions require the model to correctly identify specific persons to provide right answers, such as inquiring about someone's states, actions, attributes, positions, etc. Consider more angles for asking questions. The questions should involve ID awareness of models. For example, people in the test image have different actions, so you can ask about someone's action to evaluate whether the model recognize the identity correctly.
>
> Annotate Answers for Q&A and Descriptive Types of Questions: Answers to Q&A types of questions need to be accurate based on the character. Descriptive questions only require a correct description, as detailed as possible. Besides each person's action and state, include some simple background descriptions as well.

Table 12: The inter-annotator agreement distribution on 'Q&A' sub-task. The mean similarity is 0.9106.

| Similarity | >0.9 | 0.8-0.9 | <0.8 |
|---|---|---|---|
| proportion | 55.2% | 40.0% | 4.8% |

Table 13: The inter-annotator agreement distribution on 'Caption' sub-task. The mean similarity is 0.6822.

| Similarity | >0.8 | 0.7-0.8 | 0.6-0.7 | 0.5-0.6 | <0.5 |
|---|---|---|---|---|---|
| proportion | 22.5% | 28.2% | 37.3% | 9.9% | 2.1% |

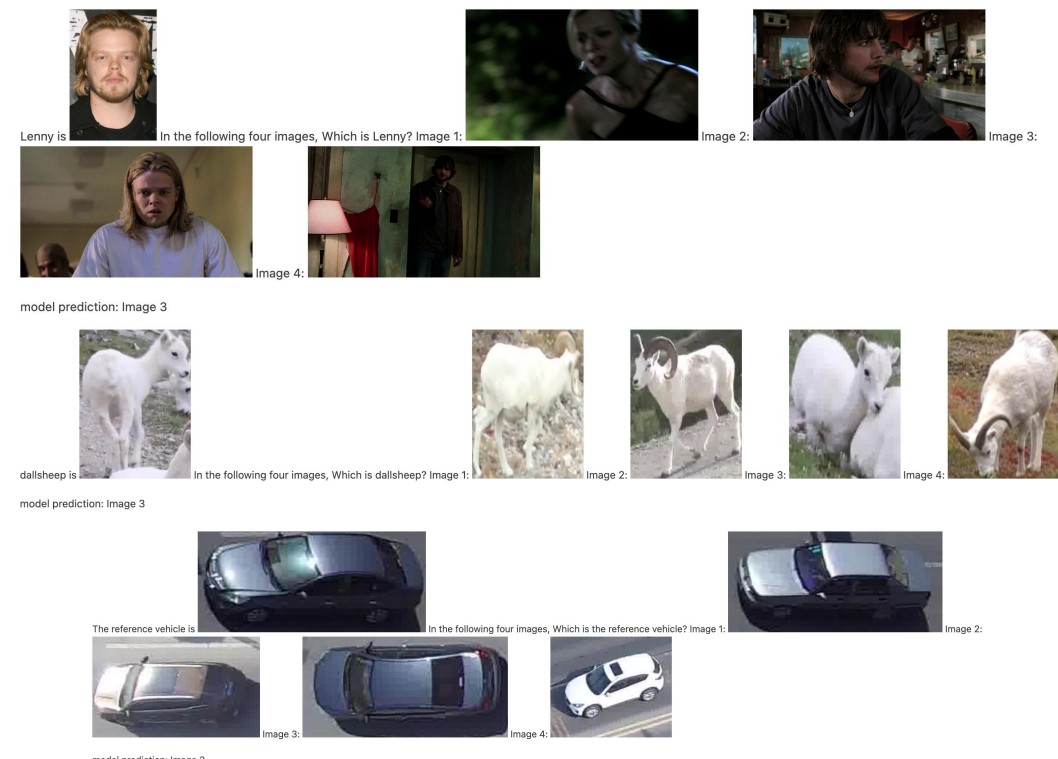

Figure 7: Samples of Matching sub-task.

Table 14: The prompts for evaluating with GPT-4 on MM-ID.

**Absolute Score**
You are a helpful and precise assistant for evaluating answers. We would like to request your feedback on the quality of an AI assistant's answer according to the given question and ground truth. The question, answer of AI assistant and ground truth will be signed by 'question', 'prediction' and 'GT'. You need to judge whether the overall meanings of prediction and ground truth answer are consistent or not. Please pay more attention to the correspondence of character names and their states or actions. Please rate the helpfulness, relevance, accuracy of the responses. You should give an overall score on a scale of 1 to 10, where a higher score indicates better overall performance. Please first output a single line containing only one value indicating the score for Assistant answer. In the subsequent line, please provide a comprehensive explanation of your evaluation, avoiding any potential bias and ensuring that the order in which the responses were presented does not affect your judgment.

**Relative Score**
We would like to request your feedback on the performance of two AI assistants in response to the user question according to the given ground truth. The question, ground truth answer and predictions of two AI assistants will be signed by 'question', 'GT', 'prediction 1' and 'prediction 2'. Please rate the helpfulness, relevance, accuracy, level of details of their responses. Each assistant receives an overall score on a scale of 1 to 10, where a higher score indicates better overall performance. Please pay more attention to the correspondence of character names and their states or actions. Please first output a single line containing only two values indicating the scores for Prediction 1 and 2, respectively. The two scores are separated by a space. In the subsequent line, please provide a comprehensive explanation of your evaluation, avoiding any potential bias and ensuring that the order in which the responses were presented does not affect your judgment.

Table 15: The prompts for closed-source APIs testing on MM-ID.

**Matching, Q&A, Caption (GPT-4V, Gemini-pro, QwenVL-Plus, QwenVL-Max)**
You are a helpful and precise assistant for providing a answer to the question. You need recognize instance identity to answer questions about reference characters or give a caption with character names (for multiple continuous images). You must provide an exact answer.

**Location (GPT-4V, Gemini-pro)**
You need to give coordinates of bounding box of some given characters or objects. The answer form should be 'bbox: [x1, y1, x2, y2].', where x1 is left side of bounding box, y1 is upper side, x2 is right side, y2 is bottom side, they are all integers.

**Location (QwenVL-Plus, QwenVL-Max)**
You need to give coordinates of bounding box of one given character or object. The answer form should only be '<ref>xxx</ref><box>(x1,y1),(x2,y2)</box>.', where x1 is left side of bounding box, y1 is upper side, x2 is right side, y2 is bottom side, they are all integers.

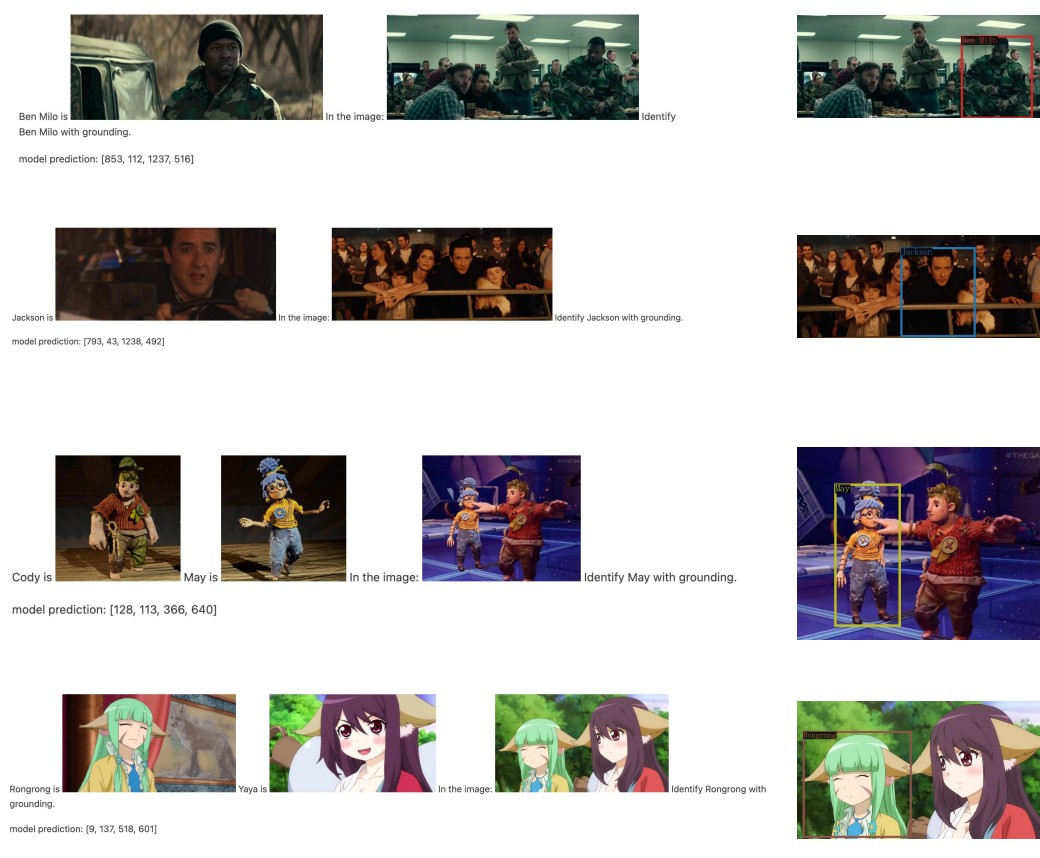

Figure 8: Samples of Location sub-task.

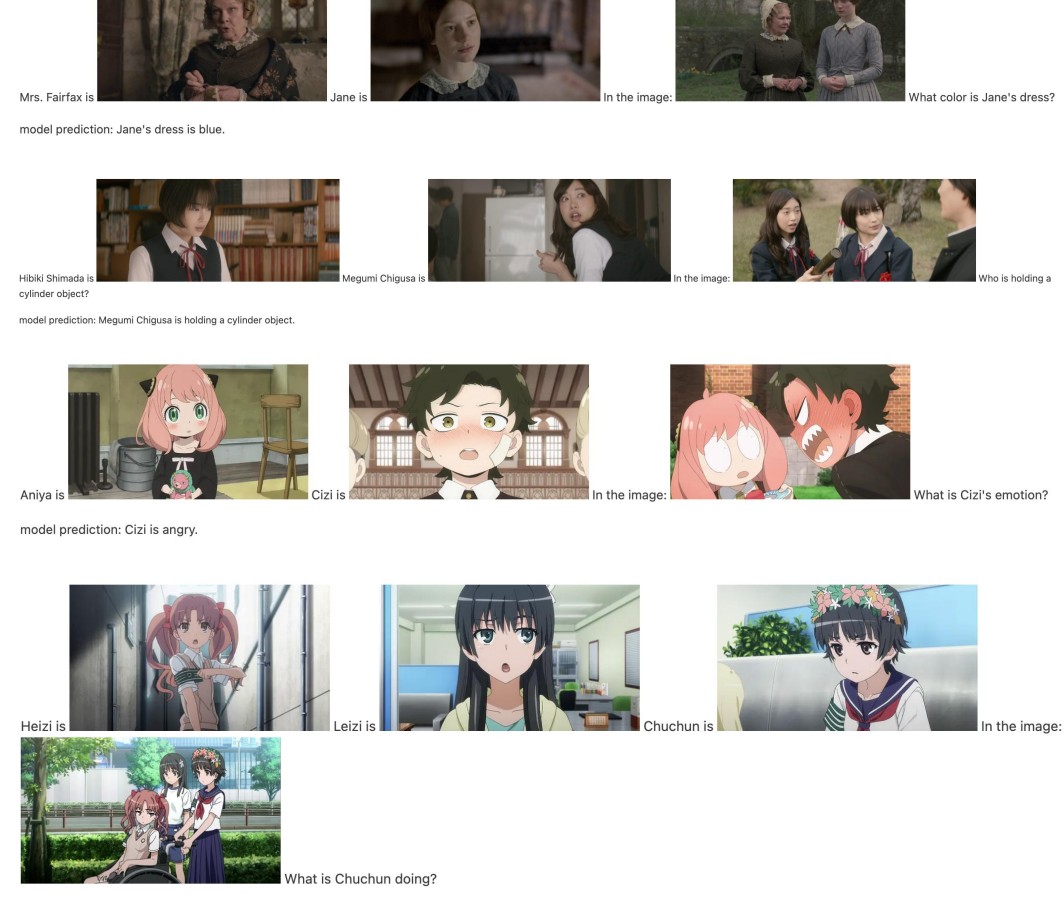

Figure 9: Samples of Q&A sub-task.

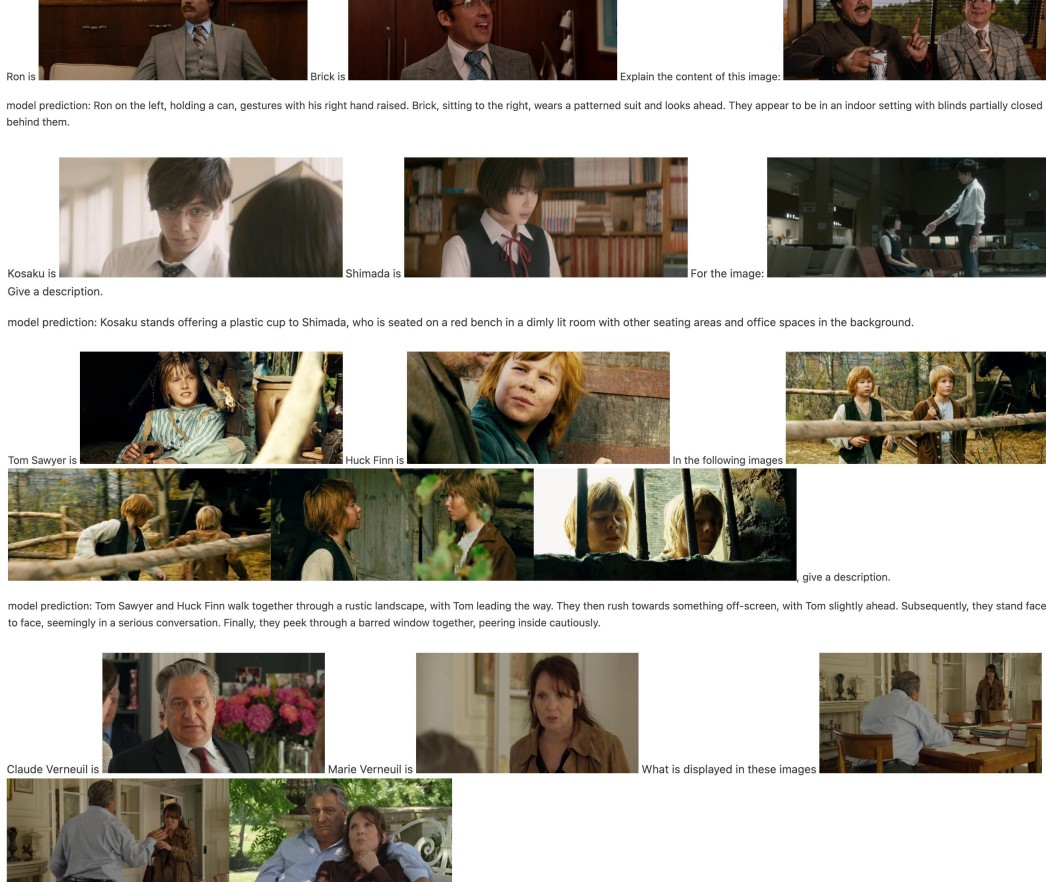

Figure 10: Samples of Caption sub-task.

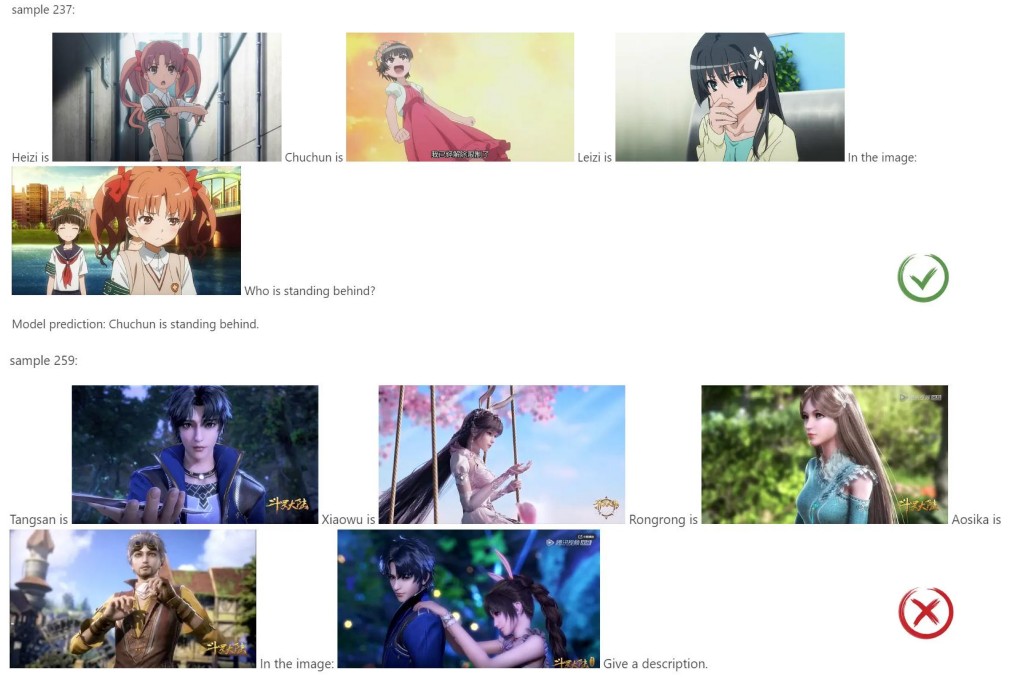

Figure 11: Samples containing additional ID images.

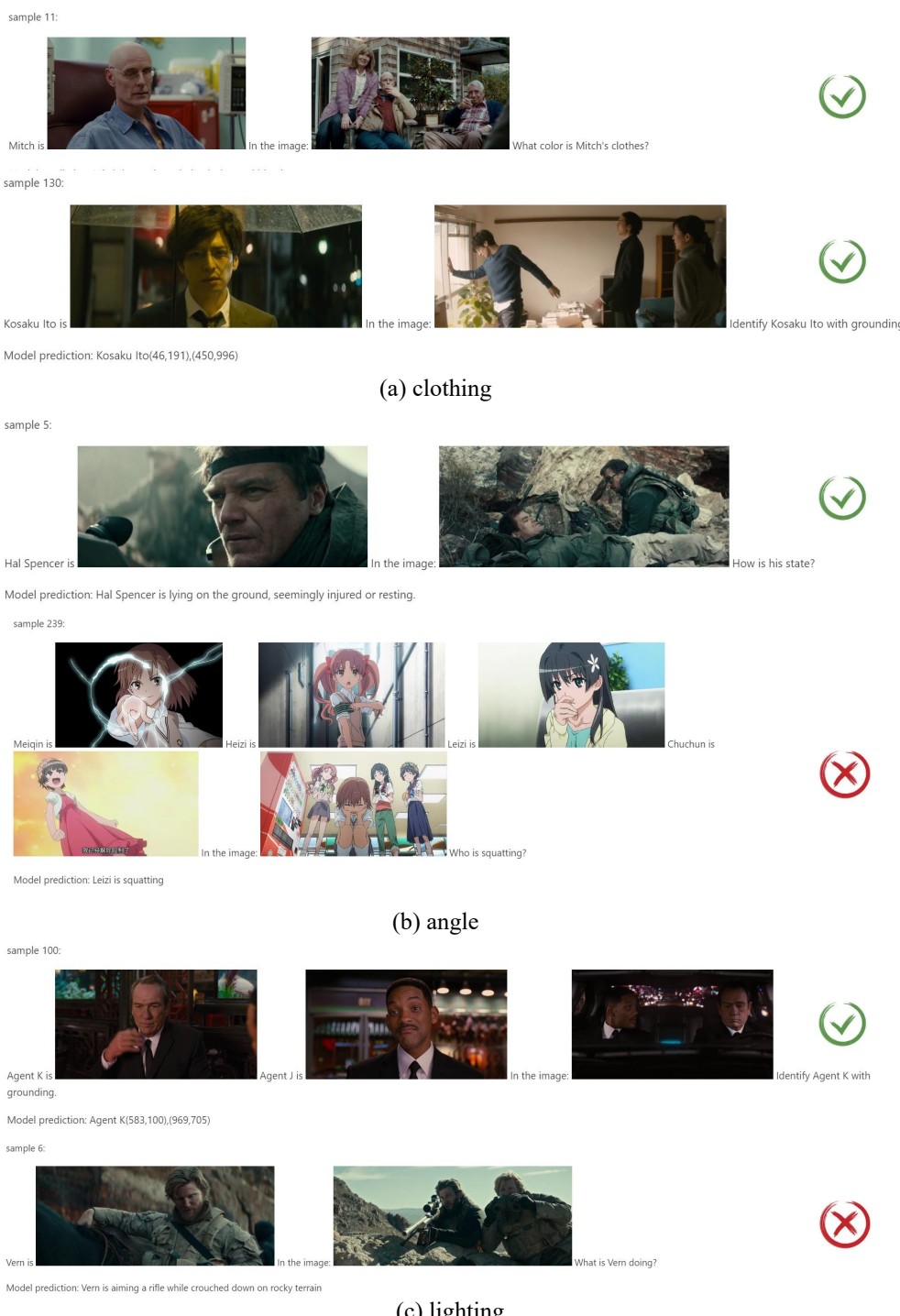

Figure 12: Model robustness to ID variations.