# OpenReview forum: "IDA-VLM: Towards Movie Understanding via ID-Aware Large Vision-Language Model"
_ICLR.cc/2025/Conference — ICLR 2025 Poster_

### Official Review · Reviewer_1ujz · 2024-10-21

**Soundness:** 4
**Presentation:** 4
**Contribution:** 3
**Rating:** 8
**Confidence:** 4

**Summary:**

The paper investigate a novel capability for large vision-language models (LVLMs) to recognize and distinguish multiple characters across different visual scenes. The authors introduce a benchmark, MM-ID, to evaluate this ability in four areas: matching, location, question-answering, and captioning. To improve LVLM performance on these tasks, they propose a tuning recipe and develop the IDA-VLM model.

**Strengths:**

- The motivation is well-grounded. The benchmark also evaluate the model’s ability to understand prompts that involve multiple images interleaved with text.
- The task of location based on reference character images is innovative. It is different from other locating tasks, such as text-to-roi grounding.

**Weaknesses:**

When constructing localization samples using RefCOCO, the ID images are cropped and flipped from the original images (this is inferred from the figure, as the text does not mention the flipping). This means that the pattern of the ID images is identical to the region of interest (ROI) in the original image. This does not align with the task defined in the paper, which aims to understand character features and locate the character in another image in a different state. The RefCOCO-based data does not require understanding of character features but simply finding identical content.

**Questions:**

- As is described in 'weaknesses', are there other ways to construct localization samples, such as using 3D datasets or leveraging base models like SAM2 and human detection models to construct samples from videos?
- When there are multiple ID images and multiple test images, how does ID-Former determine which ID image each test image should attend to? Figure 2 only illustrates the one-to-one case.

**Details Of Ethics Concerns:**

The benchmark contains
> animated images with download links collected from the internet

The copyright of these images should also be discussed in ethical considerations section.

---

> ### Author Response · Authors · 2024-11-20
>
> Thank you for your positive review and constructive suggestions.
>
> **Weakness**: ID images is identical to the region of interest (ROI) in the original image.
>
> This is true for the first stage, as mentioned in line #255, while the second-stage fine-tuning data utilizes ID images from different frames of movies, ensuring that characters appear in varying states between ID images and test images. The model need to memorize essential ID features for recognition rather than merely comparing identical content.
>
> **Question1**:  construct localization samples.
>
> These are some inspiring ways to develop localization samples, making ID images different from the corresponding characters in test images. In our work, we directly use MovieNet to construct localization samples that meet requirements.
>
> **Question2**: multiple ID images and multiple test images.
>
> When there are multiple ID and test images, concatenate and repeat queries of all ID images, then use them to conduct cross-attention with each test image. That is, the test images should attend to all ID queries. In this cross-attention, the shape of ID queries is (N_testimg, N_idimg x L_query, d), and test images' features are (N_testimg, L_img, d).
>
> **Ethics concern**:
>
> We will discuss the copyright of these downloaded images in the ethical considerations section. When releasing the dataset, we try to provide image URLs rather than directly sharing the image files.

---

> > ### Comment · Reviewer_1ujz · 2024-11-21
> >
> > I would like to thank the authors for responding. All the responses have addressed my primary concerns. I keep the score as it is.

---

### Official Review · Reviewer_JxCK · 2024-10-23

**Soundness:** 3
**Presentation:** 3
**Contribution:** 3
**Rating:** 6
**Confidence:** 4

**Summary:**

In this paper, the authors propose IDA-VLM, which is designed to conduct visual instruction tuning with ID reference. Furthermore, the authors also construct a benchmark named MM-ID to evaluate the ability of ID memory and recognition in existing  VLMs.

**Strengths:**

1.The proposed visual instruction tuning with ID reference is interesting and has the potential to be applied to movie understanding.
2.The paper is well-written and is easy to understand.

**Weaknesses:**

Problems:
1. The authors wrote: “We crop sub-images of the individuals and assign them identifiable labels, such as person names.” in lIne243. I would like to know how to assign each individual a person's name. Is the name randomly generated via Large language models e.g. GPT4?

2. Why the results of Qwen-VL-Chat are not included in Table2 and Table3. Qwen-VL-Chat should be a baseline since the proposed IDA-VLM is built based on it.

3. More models should be included in Table1. There are a lot of popular open-sourced VLMs e.g. LLaVa. These models should be included in Table1 to enhance the benchmark. and facilate future reserch.

4. In the Table4, more standard benchmarks should be included. Only evaluating IDA-VLM on two benchmarks is not enough. The authors should include more benchmarks e.g TextVQA and MMBench for a more comprehensive evaluation.


Issues:
1.The font in the figures is too small, which makes it hard for readers to read. It is highly recommended to increase the font size in figures.

**Questions:**

See Weakness

---

> ### Author Response · Authors · 2024-11-20
>
> Thank you for your valuable review and suggestions.
>
> **Weakness1**: name samples.
>
> The name is randomly sampled from a predefined set, ensuring unique names for different characters within each test image.
>
> **Weakness2**: the results of Qwen-VL-Chat
>
> In Table2 and Table3, we only compare our model with closed-sourced models, because other open-sourced models (like  Qwen-VL-Chat) have poor performance, which stems from their lack of ID-instruction learning.
>
> **Weakness3**: More models.
>
> Thanks for the suggestion. We didn't evaluate LLaVA, because the models selected for testing should support multiple images input. We will test some new open-sourced VLMs in the next version and add them to the paper later.
>
> **Weakness4**: more standard benchmarks.
>
> We currently utilize MME, SEED-Bench for evaluation, which demonstrates our model with ID instruction tuning keeps basic multimodal ability. We did not emphasize evaluation on general multimodal benchmarks (OCR, reasoning), as our primary focus was on identity-related capabilities.
>
> **Issue1**: We will increase the font size in figures.

---

> > ### Author Response · Authors · 2024-11-22
> >
> > For the weakness3, we tested LLava-Onevision, which can process multiple images. It also demonstrates weak ID aware ability, 2.86 on caption sub-task and 3.91 on QA sub-task. We added the results to new version of the paper.

---

> > ### Comment · Reviewer_JxCK · 2024-11-24
> >
> > I appreciate the responses for the authors. I still believe that the results of  Qwen-VL-Chat  should be included in Table 2 and Table 3, since the proposed method is built upon it. Am I right?

---

> > > ### Author Response · Authors · 2024-11-24
> > >
> > > Indeed, these results demonstrate the improvements of our model over the base model, Qwen-VL-Chat. Specifically, in Table 2, Qwen-VL-Chat / IDA-VLM achieves scores of 0.765 and 0.396 on QA and Caption tasks respectively. In Table 3, Qwen-VL-Chat obtains 0.052 on METEOR and 0.031 on CIDEr . We will incorporate these Qwen-VL-Chat performance metrics into both Table 2 and Table 3 for comprehensive comparison.

---

### Official Review · Reviewer_nMGR · 2024-11-04

**Soundness:** 2
**Presentation:** 3
**Contribution:** 2
**Rating:** 6
**Confidence:** 4

**Summary:**

This paper mainly focuses on VQA tasks with multiple images containing the same character identities. To reach a higher performance, the authors construct specific datasets from both (1) multi-image reasoning datasets like VCR, and (2) movie shot datasets, and utilize a certain templates to train such abilities. To evaluate such abilities, the authors propose a new benchmark MM-1D with four question types and 585 samples.

**Strengths:**

The motivation that identifies same characters in different images make sense, and it is important for VLMs.

The new benchmark MM-ID is delicately curated, with statics and detailed process to collect these data.

**Weaknesses:**

All questions in stage-2 training and MM-ID testing share a similar template as `sombody is <image>. somebody is <image> ...`. It requires the question to have single-character images and be formatted in a certain template. This is too narrow for real world usage and can hardly represent real-world performance.

Moreover, does the data used in stage-2 share the similar template. If so, how can I tell if the increase in performance is due to efficient training on such template?

The same problems also appears in the model structure in Fig. 2. It seems that the model needs to treat ID images and other images differently in the ID-Former. Moreover, I am not a little confused of the three cross-attn in ID-Former. I suppose there would be multiple ID images and multiple test images, how can I do cross attention taking multiple images as queries and keys at the same time. Please elaborate more on the model structure.

**Questions:**

My main concern is if this fixed template can cover the wide range of real-world VQA samples. Please see weakness for details.

---

> ### Author Response · Authors · 2024-11-20
>
> Thank you for your careful review and questions.
>
> **Weaknesses1**: share a similar template.
>
> This template format serves solely to inform the model about ID information, after which queries about test images can be flexible, including various questions and descriptions. The ID template functions similarly to a system prefix prompt in instruction tuning, remaining constant across interactions. Furthermore, the specific format of conveying ID information is not crucial, as we can preprocess any combination of ID role names and corresponding images into our training template.
>
> **Weaknesses2**: the increase in performance.
>
> They share the similar template. The similar templates are designed to help the model memorize the ID recognition task structure, serving as a unified format to convey ID information to the model. Performance improvements stem from our constructed instructional data that activates the model's ID perception capabilities.
>
> **Weaknesses3**: Details of the ID-Former when multiple ID images.
>
> Normal cross-attention in Q-Former uses learnable queries as Q, and image features as K, V. In contrast, the additional cross-attention (the pink one in ID-Former architecture image) in ours uses test image features as Q, and the output queries of ID images as K, V, that is, using semantic features of ID images to modulate the input embeddings of test images. When there are multiple ID and test images, concatenate and repeat queries of all ID images, then use them to conduct cross-attention with each test image. In this cross-attention, the shape of ID queries is (N_testimg, N_idimg x L_query, d), and test images' features are (N_testimg, L_img, d).
>
> **Question**: the fixed template
>
> The fixed template serves only a task format to instruct the model. In the real-world VQA samples, if you have ID role names and corresponding images, you can preprocess them into the template.

---

> ### Author Response · Authors · 2024-12-01
>
> Dear reviewer nMGR,
>
> We sincerely appreciate your review of our work and your valuable contribution to improving the quality of the paper.
>
> Have we effectively addressed your concerns in our rebuttal? If you still have any issues or new concerns, please feel free to let us know so we can continue the discussion.

---

> > ### Comment · Reviewer_nMGR · 2024-12-01
> >
> > Thanks for the authors' responses, which has clarified my confusion on the template. I agree to regard that the task is different from vanilla VQA, with IDs as additional inputs.
> >
> > As to the module structure, I am still uncertain about how ID-Former works. Accoring to your responses, image features of all ID images are concatenated in the sequence length dimension as the key & value, in order to perfom cross attention with test images as queries? If so, the authors should better draw this concatenation in the figure for better visualization.

---

> > > ### Author Response · Authors · 2024-12-02
> > >
> > > Thank you for the response and suggestion.
> > >
> > > There is a slight inaccuracy in your description. Rather than concatenating ID image features, we concatenate the output queries of the ID images from the first attention operation to serve as keys and values, in order to perfom cross attention with test images.
> > >
> > > We will draw a more detailed figure in the next version.

---

> > > > ### Comment · Reviewer_nMGR · 2024-12-02
> > > >
> > > > Thanks you for clarifying my confusion. I decide to raise the overall score.
> > > >
> > > > My opinion might be that the problem set is indeed practical, however, the solution may be somewhat straight-forward.

---

> > > > > ### Author Response · Authors · 2024-12-02
> > > > >
> > > > > Thank you for your positive feedback. We will further explore your issues and develop more flexible instruction designs for identity recognition in our future work.

---

### Official Review · Reviewer_BH7G · 2024-11-04

**Soundness:** 3
**Presentation:** 3
**Contribution:** 3
**Rating:** 6
**Confidence:** 4

**Summary:**

This paper addresses the challenge of movie understanding in VLMs by developing mechanisms to associate instances across different scenes. The authors propose ID-Former, integrated within their ID-Aware VLM architecture, and implement visual instruction tuning with ID references. To evaluate instance ID memory and recognition capabilities across multiple visual scenarios systematically, they introduce a new benchmark called MM-ID.

**Strengths:**

1) The framework architecture is conceptually straightforward.
2) The training dataset construction methodology leverages existing open-source databases with replicable procedures.
3) The MM-ID benchmark provides comprehensive evaluation metrics across multiple aspects of ID-related movie understanding tasks.

**Weaknesses:**

1. The ID-Former is the essential innovative module in the framework. The authors are encouraged to provide the architectural details and hyperparameter configurations of the ID-Former.

2. The integration of LLaVA and ShareGPT4V data. 1) The selection criteria for the 10\% subset of LLaVA data remain unclear. 2) The paper would benefit from an ablation study showing "IDA-VLM (w/ LLaVA data only)" to isolate the impact of the newly constructed tuning data.

3. The experimental setup primarily addresses scenarios where reference ID count equals or is less than the character count in test images. However, real movie scenes often involve more main and important supporting characters than visible in a single frame. Usually, it is unlikely to accurately know the characters of interest for each requested frame beforehand. Therefore, the model's performance in scenarios with excess reference IDs is worth an investigation.

4. The absence of a systematic evaluation of robustness to ID variations. Particularly, it would be interesting to know its performance regarding environmental factors (lighting, viewing angles) and character appearance changes (clothing, age progression), and whether there is a degradation under significant reference-test image discrepancies.

**Questions:**

1. The potential scoring variance inherent in subjective assessments with wide scoring ranges by GPT-4. The paper uses GPT-4 to score predictions based on a ten-point scale. Although it adopts the average of 5 times predictions, it would be nice to know whether GPT-4 can consistently provide reasonable explanations for its scoring and whether it can discriminate adjacent score points on the ten-point scale.

2. Regarding Table 5, the observed degradation in ID matching performance with first-stage tuning calls for investigation into potential mitigation strategies to maintain performance at the same level as the "w/o first stage tuning" configuration.

I am willing to change the score if my concerns are addressed.

**Details Of Ethics Concerns:**

The MovieNet dataset used in training may be subject to copyright issues.

---

> ### Author Response · Authors · 2024-11-20
>
> Thank you for your in-depth review and questions.
>
> **Weaknesses1**: Details of the ID-Former.
>
> Compared to general Q-Former, our ID-Former adds an additional cross-attention module, which is the pink one in ID-Former architecture image (Figure 2). Normal cross-attention in Q-Former uses learnable queries as Q, and image features as K, V. In contrast, the additional cross-attention in ours uses test image features as Q, and the output queries of ID images as K, V, that is, using semantic features of ID images to modulate the input embeddings of test images. When there are multiple ID and test images, concatenate and repeat queries of all ID images, then use them to conduct cross-attention with each test image.
>
> **Weaknesses2**: The integration of LLaVA and ShareGPT4V data.
>
> Due to the space limitation of the main paper, we put the ablation about the mixing rate of LLaVA and ShareGPT4V data in Appendix B (Table 10), as indicated in line #523. We will try to adjust the space and move the ablation to the main paper.
>
> The results of "IDA-VLM (w/ LLaVA data only)" may be close to results of Qwen-VL-Chat, because LLaVA data is general multimodal instruction tunign data, which can't unleash the ID-awareness ability of LVLM. Moreover, without ID instruction tuning data, IDA-VLM can't optimize ID-Former, so the result model will be similar to the baseline, Qwen-VL-Chat.
>
> **Weaknesses3**: The experimental setup primarily addresses scenarios where reference ID count equals or is less than the character count in test images.
>
> This is an interesting question, because on the one hand, characters of interest may not be in the test images. On the other hand, additional ID images may bring interference and make ID recognition harder. Actually, when we construct MM-ID benchmark, we also consider this problem to make more diverse test sampels. For example, in some questions of animation, we give more ID images than characters appearing in the test image. We added such examples of MM-ID in the new version of Appendix. In some easy Q&A samples, IDA-VLM can give right answers, but in caption sub-task, IDA-VLM may generate content of non-existing characters. This is bacause in our training data, the number of ID images generally equals to the character count in test images. If we add additional ID images to training samples, this issue can be alleviated.
>
> **Weaknesses4**: A systematic evaluation of robustness to ID variations.
>
> This is an interesting and meaningful question. In our test samples, the ID images are clear, enabling the model to capture complete identity characteristics. The test images exhibit various ID variations, including pose and clothing changes, demonstrating our model's robustness to identity variations. We included detailed case studies of ID robustness in the appendix, analyzing model performance under different ID variations.
>
> We observe that the model demonstrates robust performance against variations in clothing and viewing angles. The primary challenge arises from interference by IDs with similar features in the test images.

---

> ### Author Response · Authors · 2024-11-20
>
> **Question1**: The potential scoring variance inherent in subjective assessments with wide scoring ranges by GPT-4.
>
> Empirically, GPT-4's scoring aligns well with human subjective assessments. The following examples demonstrate GPT-4's scoring capability, where it provides reasonable explanations for its assessments while maintaining good discriminative ability across adjacent scores.
>
> Example1: consistently provide reasonable explanations
>
> > "prediction": "Caption Mitch is standing with his arms crossed.",
> "GT": "Caption Mitch is standing, crossing his arms and looking at others.",
>
> > "review1": "8\n\nThe prediction and the GT are largely consistent in conveying the main action that Captain Mitch is performing - standing with his arms crossed. Though the GT provides a bit more detail by including that he is also looking at others, this additional information does not change the overall action significantly. The slight difference is considered a minor detail and does not impact the understanding of Caption Mitch's posture or demeanor in a substantial way. The main action (standing with arms crossed) is captured correctly in both responses. Hence, the prediction is relevant, fairly accurate, and could be helpful, missing only the extra detail of where Captain Mitch is looking."
>
> > "review2": "9\n\nThe prediction and the GT are consistent in describing Caption Mitch's posture and action. Both state that Caption Mitch is standing with his arms crossed. The GT adds that he is looking at others, which is a minor detail not mentioned in the prediction but does not fundamentally change the understanding of Caption Mitch's stance or action. Thus, the additional detail in the GT is not substantial enough to significantly affect the quality of the answer. Therefore, the prediction is largely accurate and relevant, earning a high score. However, due to the slight omission of what Caption Mitch is looking at, a perfect score is not warranted."
>
> Example2: discriminative ability across adjacent scores.
>
> > "GT": "Barbara, Violet, Mattie are in a living room and discussing intensely. Barbara is standing and the other two sitting. They all hold some cards in their hands and look at them.",
>
> > "prediction": "Barbara stands in a room holding some papers, facing away from the camera. Mattie sits to the right, looking at papers in her lap. Violet sits to the left, also looking at papers.",
> > "score": "6",
>
> > "prediction": "Barbara stands in a room holding some papers, facing away from the camera. Mattie sits to the right, holding papers and looking down. Violet sits to the left, also holding papers and looking down. They appear to be in a discussion.",
> > "score": "7",
>
> **Question2**: ID matching performance.
>
> The decline in matching performance with first-stage fine-tuning can be attributed to the absence of matching tasks in the first-stage dataset, resulting in a dilution of matching-related training data. Better matching performance could be achieved by adjusting the proportion of task-specific data.

---

> > ### Comment · Reviewer_BH7G · 2024-11-27
> >
> > I appreciate the authors' response, which addresses most of my concerns. Regarding Weakness 1, the authors are encouraged to include a table that clearly outlines the parameters of the ID-Former architecture.
> > I raised the 'contribution' score and kept the other scores as they were.

---

> > > ### Author Response · Authors · 2024-11-27
> > >
> > > Thank you for your positive feedback. We appreciate your suggestion regarding Weakness 1. We will add a table to clearly outline the ID-Former architecture.

---

### Meta-Review · Area_Chair_q81T · 2024-12-19

**Metareview:**

This paper explores understanding complex visual content, such as movies with multiple characters and intricate plots. It proposes a visual instruction tuning approach for Vision-Language Models (VLMs) that incorporates identity awareness.

The reviewers praise the paper for addressing an important topic, its well-motivated and interesting method, the value of its benchmark, and clear paper writing. In the end, Most primary concerns were resolved during the rebuttal. All four reviewers are in favor of accepting the submission. The remaining issues include clarifying model parameters and the straightforward nature of the method.

The AC thinks that being straightforward yet effective does not diminish the paper's contribution. The AC recommends accepting this submission.

**Additional Comments On Reviewer Discussion:**

All four reviewers responded to the authors' rebuttal, acknowledging that their concerns had been addressed, with none opposing the acceptance of this submission.

---

### Decision · Program_Chairs · 2025-01-22

Accept (Poster)